# *Glochidion wallichianum* Leaf Extract as a Natural Antioxidant in Sausage Model System

**DOI:** 10.3390/foods11111547

**Published:** 2022-05-25

**Authors:** Chantira Wongnen, Naiya Ruzzama, Manat Chaijan, Ling-Zhi Cheong, Worawan Panpipat

**Affiliations:** 1Food Technology and Innovation Research Center of Excellence, School of Agricultural Technology and Food Industry, Walailak University, Nakhon Si Thammarat 80160, Thailand; chantira.wo@mail.wu.ac.th (C.W.); pworawan@wu.ac.th (W.P.); 2Department of Pharmacy, International Islamic University Chittagong, Kumira 4318, Bangladesh; unaiyar293@gmail.com; 3Zhejiang-Malaysia Joint Research Laboratory for Agricultural Product Processing and Nutrition, College of Food and Pharmaceutical Science, Ningbo University, Ningbo 315211, China; cheonglingzhi@nbu.edu.cn

**Keywords:** meat, lipid oxidation, plant, phenolic, antioxidant

## Abstract

This study highlighted the role of an 80% ethanolic Mon-Pu (*Glochidion wallichianum*) leaf extract (MPE), a novel natural antioxidative ingredient, in controlling the oxidative stability and physicochemical properties of a cooked sausage model system (SMS). MPE had a total extractable phenolic content of 16 mg/100 g, with DPPH^●^ scavenging activity, ABTS^●+^ scavenging activity, and ferric reducing antioxidant power of 2.3, 1.9, and 1.2 mmole Trolox equivalents (TE)/g, respectively. The effects of different concentrations of MPE (0.01–10%, *w*/*w*) formulated into SMS on lipid oxidation, protein oxidation, and discoloration were compared to synthetic butylated hydroxyl toluene (BHT; 0.003%, *w*/*w*) and a control (without antioxidant). The peroxide value (PV), thiobarbituric acid reactive substances (TBARS), and protein carbonyl contents of SMS tended to increase with increasing MPE concentration (*p* < 0.05), indicating that high MPE excipient has a pro-oxidative effect. The lowest lipid oxidation (PV and TBARS) and protein carbonyl contents were observed when 0.01% MPE was used to treat SMS (*p* < 0.05), which was comparable or even greater than BHT-treated SMS. High concentrations (1–10%) of MPE incorporation led to increases in the discoloration of SMS (*p* < 0.05) with a negligible change in pH of SMS. The water exudate was reduced when MPE was incorporated into SMS compared to control (*p* < 0.05). Furthermore, MPE at 0.01% significantly reduced lipid oxidation in cooked EMS during refrigerated storage. According to the findings, a low amount of MPE, particularly at 0.01%, in a formulation could potentially maintain the oxidative stability and physicochemical qualities of cooked SMS that are comparable to or better than synthetic BHT.

## 1. Introduction

Because lipids have such a large impact on food quality, lipid oxidation is one of the most essential features of shelf life assessments. Most processed meat products’ shelf life is governed by lipid deterioration, particularly in products high in lipids and polyunsaturated fatty acids [1,2,3,4,5]. Examples include fresh sausage, cooked (uncured) meats, and dried sausage. Pork and poultry, which have high unsaturated fat content, are particularly sensitive [6]. The rancid flavor is one of the most objectionable flavor components of meat lipid oxidation, and it impairs the sensory quality of the finished product. Lipid oxidation in fat-based food products produces free radicals or reactive oxygen species (ROS) such as peroxyl, hydroxyl, and alkoxyl free radicals, as well as other products such as malondialdehyde, acrolein, 4-hydroxy-trans-hexanal, 4-hydroxy-trans-nonenal, and crotonaldehyde-like substances, which are purportedly linked to aging, carcinogenesis, mutagenesis, atherosclerosis, diabetes, inflammation, and arthritis [1,2,7,8,9]. To counteract oxidative damage, antioxidants are incorporated in all processed meats. Antioxidants, both endogenous and dietary, may help to mitigate the negative effects of free radicals in muscle-based food systems [10]. Cured meats contain sodium nitrite, which is a very effective antioxidant, and in some countries, the synthetic phenolic antioxidants butylated hydroxy anisole (BHA) and butylated hydroxy toluene (BHT) are commonly used in uncured meats [6,11]. According to USDA guidelines, BHA and BHT are allowed up to 0.01% (based on fat content) in fresh sausage and 0.003% (based on total weight) in dry sausage [6]. According to Commission Regulation (EU) No. 1129/2011 [10], antioxidants such as ascorbic acid and its salts, phosphoric acid, calcium disodium ethylene diamine tetraacetate (calcium disodium EDTA), and rosemary extracts are permissible in heat-treated processed meat in European Union countries. However, because synthetic antioxidants are causing worry among consumers, numerous meat processors are exploring natural antioxidant alternatives.

Natural antioxidants have lately gained popularity as a result of their increased safety and consumer perception, as well as their potential for disease prevention and health promotion [7,12]. In multiple studies, the antioxidant activity of various fruits, vegetables, and herbs was found to be strongly linked to their overall phenolic content. Many plant extracts have been shown to have antioxidation properties when used in oils, fats, and fat-containing foods, as well as meat products [6,13,14]. Plant secondary metabolites, notably phenolics, have been reported to have better antioxidant power than synthetic antioxidants such as BHA and BHT in numerous investigations [15,16]. In chilled and pre-cooked frozen sausages, commercial rosemary extract (RE; 2500 ppm) is as effective as BHA/BHT. RE, on the other hand, was found to be more effective than BHA/BHT at lowering thiobarbituric acid reactive substances (TBARS) levels and preventing red color loss in uncooked frozen sausage [6].

Polyphenols, also known as phenolic compounds, are a class of molecules made up of an aromatic benzene ring substituted by hydroxyl groups, and are capable of a variety of bioactivities such as antioxidant activity [7,17,18]. Antioxidant action is connected to the structure of phenolic compounds, which is regulated by the number and placement of hydroxyl groups, as well as various substitutions and glycosylation of the molecule [17,19,20]. As a result, the antioxidant effects of plants containing phenolic compounds are linked to phenolic components in plants and are influenced by chemical structure. The redox characteristics of phenolic compounds, which can aid in the absorption and neutralization of free radicals, the quenching of singlet and triplet oxygen, and the decomposition of peroxides, are primarily responsible for their antioxidant activity [21]. Antioxidant compounds can minimize the oxidation of lipids or other biomolecules by inhibiting the onset or propagation of oxidative chain reactions [21]. Several plant polyphenolic compounds have been found as antioxidants or free radical scavengers with health advantages [22]. Higher total phenolic content leads to increased oxidative stability and effectively protects fresh chicken sausages against lipid oxidation during frozen storage [23]. Plant phenolics can increase the oxidative stability of sausages during storage, according to Yu et al. [24], and phenolics have greater phospholipid oxidation exhibition potential than BHT at the end of storage (120 days).

*Glochidion wallichianum* Mull. Arg., commonly known as Mon-Pu (Thai name; MP; Figure 1), is a Euphorbiaceae shrub to tree found in the evergreen forests of Thailand, Burma, Malaysia, and South China [25]. MP is an edible plant native to Southern Thailand, where its young leaves are eaten raw and prepared in the same way as other local vegetables. MP, like other edible indigenous plants, is strong in antioxidant components that help prevent diseases such as cancer, neurological disease, and cardiovascular disease [26]. The alcoholic leaf extracts of MP have high phenolic content and robust antioxidative activity [25]. MP leaf extract (MPE) was discovered to have antioxidant activities by Alzoreky and Nahakara [27]. According to the ferrylmyoglobin/2,20-azino-bis-3-ethylbenzthiazoline-6-sulphonic acid radical (ABTS^●+^) scavenging experiment, 80% methanolic extract shows good antioxidant capability and a high level of phenolic compounds. Panpipat et al. [7] also discovered that MPE prepared by 50% acetone and 80% methanol contained significant phenolic content, which might boost 2,2-diphenyl-1-picrylhydrazyl radical (DPPH^●^) and ABTS^●+^ inhibitory activity, as well as ferric reducing antioxidant power (FRAP). Tangkanakul et al. [28] also demonstrated that the 95% ethanolic MPE had stronger DPPH^●^ inhibition and phenolic content than the aqueous extract. Furthermore, due to its antioxidant capacity, MPE has been utilized to reduce post-harvest melanosis in shrimp [29]. MPE is applied as an herbal medicine, an antimicrobial agent, and in cosmetics in addition to food [21,30]. The active antioxidant elements of MPE include beta-carotene, lutein, polyphenolic compounds, triterpenoids, vitamin E, and vitamin C [25,26]. The most abundant antioxidant ingredient in MPE is total polyphenol, particularly gallic acid, which is connected to its considerable antioxidant activity [30,31]. Anantachoke et al. [25] isolated gallic acid, methyl gallate, luteolin-8-C-β-D-glucopyranoside (orientin), luteolin-6-C-β-D-glucopyranoside (isoorientin), and apigenin-8-C-β-D-glucopyranoside (vitexin) from methanolic MPE.

Overall, MPE can be applied to prevent oxidative instability in meat emulsion products due to its antioxidant properties in vitro [7] and in an oil-in-water emulsion model [28]. Phenolics, on the other hand, are known to have both antioxidant and pro-oxidant effects on biological systems as redox-active molecules. [14,32]. Thus, the effective concentration of MPE must be optimized in specific products. The goal of this investigation was to see if MPE could serve as a natural antioxidant in a sausage model system (SMS). It can be thought of as a green, consumer-friendly method of reducing lipid oxidation in muscle-based foods.

## 2. Materials and Methods

### 2.1. Chemicals

All chemicals and reagents utilized in this study were procured from Sigma-Aldrich (St. Louis, MO, USA).

### 2.2. Preparation of MPE

Freshly harvested leaves of *Glochidion wallichianum* Mull. Arg., or Mon-Pu (Thai name; MP) (5-kg), were purchased from Thasala market, Nakhon Si Thammarat, Thailand (Figure 1). To prevent or minimize the decomposition and/or oxidation of active compounds found in the leaves, leaves were dried at 60 °C for 72 h in a hot air oven until the moisture content was <10% [7]. Dried samples were ground to a fine powder using a grinder (Panasonic MK-G20MR, Japan) and passed through a 25-mesh sieve. The ground samples were vacuum-packed and stored at −40 °C until required for phenolic extraction.

Dried MP powder was extracted with 20 vol of 80% ethanol at 50 °C for 15 h in a shaking water bath [7,12]. Following that, the mixture was filtered using Whatman No. 4 filter paper (Whatman International Limited, Kent, UK). To remove the solvent, the filtrate was evaporated at 40 °C using a rotary evaporator. The residual solvent was flushed with N_2_ and the solvent-free extract was subjected to freeze-drying. Total extractable phenolic content, free radical scavenging activities, and reducing power were all determined after the freeze-dried MPE was pulverized and sieved through a 25-mesh sieve. The dried MPE was keep frozen at −40 °C under vacuum until it was required for application in SMS.

### 2.3. Determination of Total Extractable Phenolic Content, Free Radical Scavenging Activities, and Reducing Power

Total extractable phenolic content was estimated using the Folin–Ciocalteu colorimetric method [33]. Total extractable phenolic content was reported as mg of gallic equivalents (GAE)/gram. The DPPH^•^ and ABTS^•+^ scavenging activities were assessed according to Sungpud et al. [34] and Panpipat et al. [7], respectively. FRAP assay was used to determine the reducing capacity of MPE [35]. For all assays, Trolox (0–1 mM) was used as a standard. The results are reported as mmole Trolox equivalents (TE)/gram.

### 2.4. Preparation of SMS

SMS was prepared by chopping 73.2% (*w*/*w*) minced pork loin (*Longissimus dorsi*) with 2.5% (*w*/*w*) salt for 3 min and then adding 24.3% (*w*/*w*) soybean oil and subsequently chopping for 4 min, using a Talsa Bowl Cutter K15e (The Food Machinery Co., Ltd., Kent, UK). The MPE with different concentrations (0%, 0.01%, 0.1%, 1%, and 10% *w*/*w*) or 0.003% BHT [6], based on total weight of SMS, were added and further chopped for 3 min. After stuffing into the casing (2 cm diameter), two-step cooking (60 °C/30 min followed by 90 °C/30 min) was applied. The product’s core temperature was no less than 72 °C to ensure food safety. After cooling in ice water for 10 min, the samples were randomly taken for analyses for lipid oxidation, protein oxidation, discoloration, pH, and drip loss. The selected formulation and the control SMS were packed individually in polyethylene bags and kept refrigerated for 7 days (4–5 °C). The samples were obtained on days 0, 3, 5, and 7 for lipid oxidation analysis.

### 2.5. Determination of Lipid Oxidation

Bligh and Dyer’s method [36] was used to extract lipid from the samples. According to Panpipat et al. [37], the peroxide value (PV) and TBARS were determined. The PV is reported as milliequivalents (meq) of free iodine/kg sample. TBARS is reported as mg malondialdehyde (MDA) equivalent/kg sample.

### 2.6. Determination of Protein Oxidation

Protein carbonyl content was determined as an index of protein oxidation [38]. For 1 h at room temperature (27–29 °C), 2.0 mL of 10 mM 2,4-dinitrophenylhydrazine (DNPH) in 2 M HCl was reacted with 0.5 mL of 4 mg/mL protein solution. To precipitate protein after incubation, 2 mL of 20% (*w*/*v*) trichloroacetic acid was added. To separate unreacted DNPH, the pellet was washed twice with 4 mL of ethanol:ethylacetate (1:1, *v*/*v*), blow-dried, and dissolved in 1.5 mL of 0.6 M guanidine hydrochloride in 20 mM potassium phosphate (pH 2.3). At 370 nm, the protein’s absorption was measured and the protein carbonyl content was estimated using a molar absorptivity of 22,400 M^−1^ cm^−1^.

### 2.7. Determination of Redness Index and Discoloration

The *a* * and *b* * values of the cooked SMS were determined using a Hunterlab ColorFlex^®^EZ instrument (Hunter Assoc. Laboratory; Reston, VA, USA), equipped with an illuminant D65 and 10° observer light source. The following formula was used to determine the redness index [39]:(1)$$\mathrm{Redness}\;\mathrm{index}=\;\frac{a{\;}^{*}}{b^{\;*}}$$

The discoloration percentage of the sample was calculated against the redness index of the control (without antioxidant).

### 2.8. Determination of pH and Expressible Drip

A pH meter (Cyberscan 500, Eutech, Singapore) was used to determine the pH of the homogenate (sample:distilled water, 1:10). For the expressible drip, a sample was weighed and sandwiched between layers (2 pieces at the top and 3 pieces at the bottom) of Whatman No. 1 filter paper. The sample was reweighed after being compressed for 2 min with the 5 kg standard weight. Expressible drip was calculated as percentage of sample weight [40].

### 2.9. Statistical Analysis

All of the tests were carried out in triplicate (*n* = 3). An ANOVA analysis was performed on the data. Duncan’s multiple range test was used to compare the means. For pairwise comparison, the t-test was used. The statistical analysis was carried out using SPSS 23.0 (SPSS Inc., Chicago, IL, USA).

## 3. Results and Discussion

### 3.1. Total Extractable Phenolic Content and In Vitro Antioxidant Activity of MPE

Plant extracts are commonly used to protect food from deterioration and spoilage by preventing lipid and protein oxidation and inhibiting microbial growth [41,42]. Due to their reducing capacity, hydrogen donor efficacy, and singlet oxygen scavenger, phenolic compounds make the most of their antioxidant and antibacterial properties [43,44]. Moniruzzaman et al. [45] discovered that phenolic compounds had a good association with DPPH^●^ inhibition and FRAP values, which represent radical inhibition and reducing power. The antioxidative activity, however, is influenced by their structure, hydroxyl group position and number, polarity, and, most importantly, the bond dissociation energy required to remove the hydrogen atom [19].

MPE has a total extractable phenolic content of 16 mg/100 g. (Figure 2). For DPPH^●^ inhibitory activity, ABTS^•+^ inhibition, and FRAP, the MPE had 2.3, 1.9, and 1.2 mmole TE/g, respectively (Figure 2). This was because the extract contained extractable phenolic components as well as other bioactive phytochemicals. Total polyphenol, notably gallic acid, was the most abundant antioxidant molecule in MPE, which is linked to its significant antioxidant activity [30,31]. Methyl gallate, vitexin, orientin, and isoorientin were also found in MPE [25].

### 3.2. Effect of MPE on Lipid Oxidation of Cooked SMS

Figure 3 depicts the inhibitory activity of different concentrations of MPE against lipid peroxidation in a cooked SMS, as represented by primary (PV) and secondary (TBARS) oxidation products. The PV of the SMS-treated 0.01% MPE and BHT was significantly lower (Figure 3a, *p* < 0.05) than that of the control (without antioxidant), whereas the PV of the SMS-treated 0.1–10% MPE was comparable to that of the control SMS (Figure 3a, *p* > 0.05). This result demonstrated the concentration limit of MPE in inhibiting lipid oxidation of SMS. The inhibitory effect of MPE towards lipid oxidation could be attributed to its polyphenolics such as gallic acid methyl gallate, vitexin, orientin, and isoorientin [25,30,31] and other bioactive phytochemicals such as beta-carotene, lutein, polyphenolic compounds, triterpenoids, vitamin C, and vitamin E [25,26]. The polyphenols are capable of inhibiting radical formation and the propagation of reaction chains during the oxidation process [5,7]. The phenolic antioxidants found in MPE can effectively scavenge free radicals in SMS when used in a low concentration due to their excellent dispersion in emulsion systems. MPE may have quenched both hydrophobic and polar free radicals, as evidenced by DPPH^●^ and ABTS^•+^ scavenging capabilities (Figure 1). The lowest TBARS value was found in SMS treated with 0.01% MPE, followed by 0.1% MPE and BHT (*p* < 0.05), with a non-significant difference between the latter two. It should be noted that adding 1–10% MPE resulted in increased TBARS values in SMS that were comparable or even greater than the control sample (Figure 3b, *p* < 0.05). The polyphenol concentration governs the retardation of lipid oxidation in MPE-treated SMS, and excessive phenolics are detrimental to the improvement of the SMS antioxidant activity. This was consistent with Cheng et al. [46], who noticed a remarkable upward trend in inhibiting lipid oxidation as mulberry polyphenolic addition increased from 0% to 1.0%, followed by a rapid decline once the addition exceeded 2.5%, with the highest lipid oxidation at 10% mulberry polyphenols added in myofibrillar protein emulsion. It could be explained that excessive polyphenols interacted with myofibrillar protein, causing a change in myofibrillar protein molecular structure at the interface. A high concentration of phenolics may also serve as a deterrent, preventing the formation of a continuous protein layer around oil droplets, thereby preventing the formation of high-elastic emulsion [46]. The high dose of phenolic compounds promotes the production of amino-quinone and thiol-quinone adducts, which may prevent intermolecular interactions from forming between proteins at the emulsion interface. The disordered emulsion may also be able to react with atmospheric oxygen, stimulating lipid oxidation in SMS treated with high MPE concentrations. Aside from antioxidative activity, certain phenolic excipients may actively contribute to protein unfolding and denaturation, leading to a significant reduction in the antioxidant capacity of the MPE in SMS [47]. The percentage lipid oxidation retardation of MPE-treated SMS at various concentrations calculated from PV and TBARS was consistent with the PV and TBARS values (Figure 3c). The 0.01% MPE-treated sample had the highest inhibitory activity toward primary and secondary lipid oxidation products, outperforming BHT particularly in limiting aldehyde formation. It should be noted that the addition of 1% MPE had no effect on lipid oxidation (Figure 3c). Based on the results, a low dose of polyphenols improves the antioxidant activity of MPE.

### 3.3. Effect of MPE on Protein Oxidation of Cooked SMS

During SMS processing, alkaline amino acids (lysine, arginine, and proline) from muscle proteins are oxidatively modified, resulting in the formation of the protein carbonyl via the creation of a carbon-centered protein radical in the existence of ROS, followed by oxidative deamination facilitated by transition metals [48]. Because iron and myoglobin have been demonstrated to be excellent promoters of carbonyl formation in myofibrillar proteins [49], they could have contributed mostly to protein carbonylation examined in this investigation. The occurrence of protein carbonyl in SMS treated with different concentrations of MPE compared to BHT-treated SMS and control SMS is depicted in Figure 4. Increasing the MPE concentration in SMS from 0.01% to 10% resulted in a substantial increment in protein carbonyl content (*p* < 0.05). The sample treated with 0.01% MPE had the lowest protein carbonyl content (*p* < 0.05), which was superior to synthetic BHT-treated SMS. The 0.1% MPE-formulated SMS had comparable inhibitory activity against protein oxidation to the BHT-treated sample (*p* > 0.05). It should be noted that the addition of 1–10% MPE to SMS resulted in higher protein carbonyl content than the control sample (*p* < 0.05), indicating that high amounts of MPE stimulate protein oxidation. This is consistent with Cando et al. [14], who investigated the significant pro-oxidant effect of Willowherb extract when added to beef patties at high concentrations (800 ppm). The pro-oxidative functions of polyphenols have been attributed to their activity to convert molecular oxygen to ROS and Fe^3+^ to Fe^2+^ [50]. The increase in ROS formation and/or support to the iron oxidizing cycle may theoretically influence both lipids and proteins. As a result, the increase in protein carbonylation mediated by MPE phenolics could be explained by the particular interaction mechanisms between phenolics and specific protein residues in dietary proteins [49]. According to this pathway, the δ-amino group of alkaline amino acids (lysine, arginine, and/or proline) can interact with quinone derivatives of phenolics formed in the presence of transition metals, initiating the oxidative deamination of the amino acid and eventually generating the respective carbonyl compound [49]. It is thus possible that an undetermined proportion of the phenolics in the MPE were oxidized in the presence of endogenous iron, and the resulting quinones catalyzed the formation of protein carbonyls due to their deamination activity on alkaline amino acids [51]. As a result, the quinone forms would have aided oxidative deamination of alkaline amino acids, as well as lipid oxidation. The extent of carbonyl content was 2–14 nmole/mg protein as the limit of protein oxidation [52]. In this study, the carbonyl content of all MPE-treated samples ranged from 0.58 to 1.40 nmole/mg, and the BHT-treated sample was 0.80 nmole/mg protein, but the control sample was 1.10 nmole/mg protein. Protein oxidation in SMS as affected by MPE concentration highlights a clear antioxidant/pro-oxidant effect in relation to lipid counterparts. Plant phenolics, as redox-active compounds, have been reported to have both pro-oxidant and antioxidant effects on biological systems [32], relying on the availability of other redox-active substances and transition metals [50]. The antioxidant activity of polyphenols is dependent on both ROS inhibition and chelation activity [14]. These results indicate that 0.01–0.1% MPE addition effectively reduces protein oxidation in SMS.

### 3.4. Effect of MPE on Redness Index and Discoloration of Cooked SMS

Figure 5 depicts the effect of MPE on the redness index of cooked SMS. The color of SMS was unaffected by 0.01% MPE and BHT, and the redness index was identical to the control (*p* > 0.05). The lower dose of MPE (0.01%) was already as effective as the synthetic antioxidant BHT (0.003%), and increasing the concentration of phenolics provided no additional benefit. Gallic acid (100 ppm), one of the major phenolics in MPE, was discovered to be an effective lipid oxidation inhibitor in beef patties, as well as a protective factor against oxymyoglobin oxidation [53]. The redness index increased (*p* < 0.05) when the MPE was raised up to 1%. The elevated redness index was caused by the MPE’s activity to maintain the redox state of myoglobin in the cooked SMS. However, the redness index decreased (*p* < 0.05) when the MPE was added at a 10% concentration. The darkening effect of MPE at high concentrations, where MPE had a brownish tint, was probably responsible for the lower redness index. It has been found that the incorporation of the Willowherb (*Epilobium hirsutum* L.) extract at 800 ppm resulted in more severe discoloration [14]. Furthermore, the oxidation of MPE, as well as the production of metmyoglobin accelerated by high MPE concentrations, may all contribute to the reduced redness index. The creation of quinone forms, which was a feasible explanation for protein carbonylation, could also explain why SMS with additional rich phenolic extracts brown faster [54]. The availability and utilization of oxygen are required for the production of quinones from plant phenolics. Browning reactions such as Maillard and enzymatic reactions, in which transition metals such as iron play a prominent role, are credited with this oxygen consumption process [14]. Low oxygen levels in the meat matrix may promote the development of metmyoglobin and, as a result, meat discoloration [53]. Metmyoglobin accumulation and meat discoloration were shown to be mostly dependent on the existence of reducing equivalents in meat and lipid oxidation [53]. The ferrous oxymyoglobin is known to be oxidized into ferric metmyoglobin by primary oxidation products such as peroxides and other ROS [53]. McBride et al. [55] found that RE delayed lipid oxidation but had no effect on the redness in ground beef. These oxidized plant phenolics are thought to hasten the oxidation of oxymyoglobin [54]. Overall, both an increase and a decrease in the redness index indicated that the cooked MPE had discolored (Figure 5). When comparing the discoloration of cooked SMS to the control group, the maximum degree of discoloration was determined to be 1% MPE (60% discoloration), followed by 10% MPE (25% discoloration), and 0.1% MPE (21% discoloration), although 0.0% MPE and BHT had no influence on cooked SMS discoloration. As a result, 0.01% MPE is considered the best concentration for preventing SMS discoloration.

### 3.5. Effect of MPE on pH and Expressible Drip of Cooked SMS

The pH of all treatments was not different, except for sample with 10% MPE, which was a bit lower than the others (Figure 6a). The results suggest that MPE at 0.01–1% can be added without any changes in pH of the SMS. A high concentration of MPE, i.e., 10%, may cause proton release to a greater extent during cooking SMS, resulting in a pH reduction of the cooked SMS. It has been reported that free phenolic acid was released during heating [56]. The expressible drip of cooked EMS incorporated with MPE at different concentrations in comparison with that added with BHT is shown in Figure 6b. All the treatments had lower expressible drip than the control (*p* < 0.05), suggesting the higher water/lipid holding capacity of the cooked SMS with MPE and BHT. However, MPE at 0.01% rendered the cooked SMS with the lowest expressible drip, suggesting the highest water/lipid retention in the matrix. MPE at a low concentration may effectively cross-link the proteins to form networks with water holding capacity (WHC) and lipid retention ability. MPE with concentration higher than 0.1% had higher expressible drip than that with 0.01% (*p* < 0.05), which may be linked to the net results of protein and lipid oxidation as well as the degree of gelation and the emulsifying ability of the MPE at different concentrations. Guo et al. [57] reported that phenolic compounds such as gallic acid, chlorogenic acid, propyl gallate, quercetin, catechin, and (−)-epigallocatechin-3-gallate (EGCG) caused unfolding and promoted cross-linking of myofibrillar proteins in myofibrillar protein–emulsion composite gel. All phenolics retarded the lipid oxidation of gel upon refrigerated storage [57].

The addition of oxidized phenolic compounds at the optimal level (0.05–0.20%, depending on types of phenolic) could increase the gel strength of bigeye snapper (*Priacanthus tayenus*) surimi [58]. For instance, in comparison to the control, gels supplemented with 0.05% oxidized tannic acid showed significant increases in both breaking force and deformation, which was linked to reduced expressible moisture content [58]. It has been reported that emulsification and gel formation of myofibrillar proteins were hindered by high levels of EGCG (500–1000 mg/L). Lower amounts of EGCG (50–200 mg/L) increased the oxidative stability of meat emulsions without compromising textural feature [59]. Quan et al. [60] reported that the interaction between proteins and polyphenols is crucial in improving the quality of some foods. The interaction between proteins and nonpolar polyphenolics, for example, could elevate the surface hydrophobicity of the proteins, improving their emulsifying characteristics. Thus, protein–polyphenol conjugates have been proposed as effective antioxidant emulsifiers that can locate and act at the oil–water interface, preventing oxidation in emulsified foods [60]. Heating also stimulates protein connections between the continuous phase and the membrane surrounding the surface of the lipid droplets during the manufacture of emulsified meat products, preventing lipid coalescence and water movement and resulting in a stable product [61]. With 0.01% MPE, the SMS had the lowest TBARS (Figure 3b). The presence of low levels of MDA may have a beneficial effect on the SMS’s water retaining capacity. Moderate alteration of myofibrillar protein by MDA is widely regarded to improve the WHC, texture, and overall quality of myofibrillar protein gel [61,62]. However, due to polymerization and aggregation of proteins, significant alteration by high amounts of MDA can reduce the gel’s overall quality [46].

### 3.6. Effect of MPE on Lipid Oxidation of Cooked SMS during Refrigerated Storage

PV and TBARS values of cooked SMS prepared with and without 0.01% MPE during refrigerated storage are shown in Figure 7a,b. Both samples’ PV and TBARS values significantly increased over time (*p* < 0.05). However, the control sample (without 0.01% MPE) showed greater PV and TBARS levels than the sample with 0.01% MPE throughout storage. From the results, the antioxidant activity of MPE in cooked SMS was validated, and MPE can help to prevent lipid oxidation in SMS during cooking and subsequent refrigerated storage. Thus, MPE at 0.01% demonstrated a carry-through property, which was demonstrated by delayed lipid oxidation after cooking and on storage [63].

## 4. Conclusions

In vitro free radical scavenging activity and reducing power were found in an MPE with extractable phenolic compounds of 16 mg/100 g. MPE is a unique natural antioxidative component that helps manage the oxidative stability and physicochemical properties of cooked SMS. Overall, a small amount of MPE in a formulation, especially 0.01% (*w*/*w*), may have delayed lipid and protein oxidation, reduced expressible drip, and prevented the discoloration of cooked SMS that is comparable to or better than synthetic BHT. Furthermore, MPE at 0.01% substantially delayed lipid oxidation in cooked SMS during refrigerated storage, indicating effective carry-through property.

## Figures and Tables

**Figure 1 foods-11-01547-f001:**
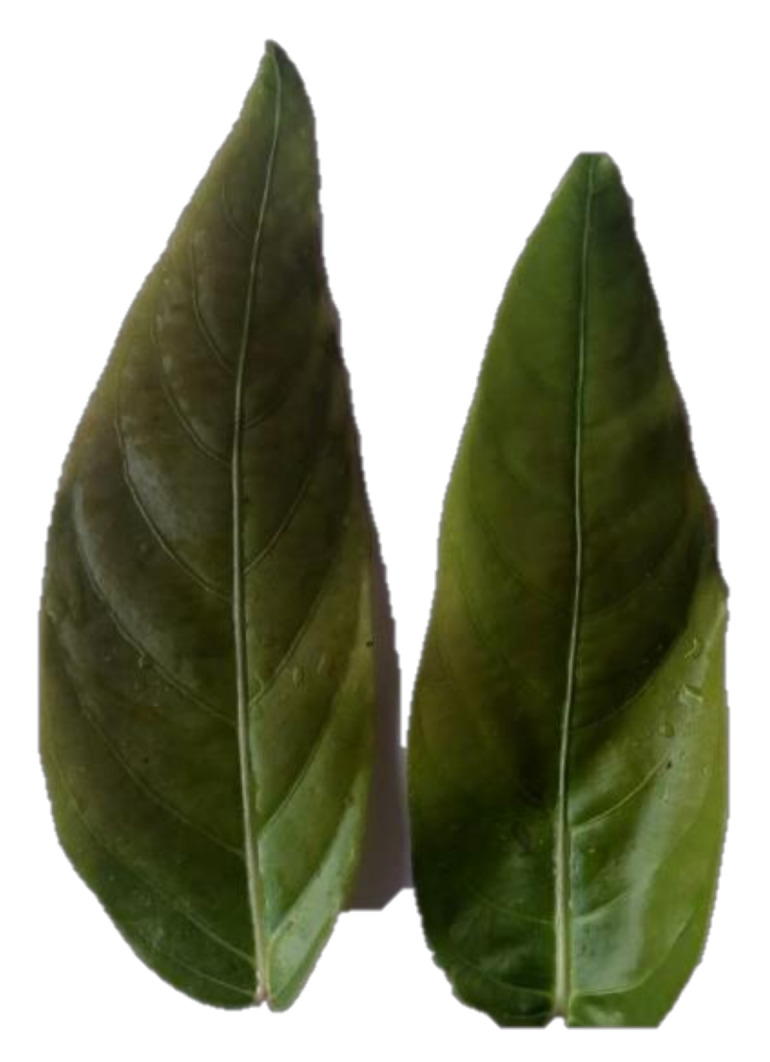
Leaves of *Glochidion wallichianum* Mull. Arg., commonly known as Mon-Pu (Thai name; MP).

**Figure 2 foods-11-01547-f002:**
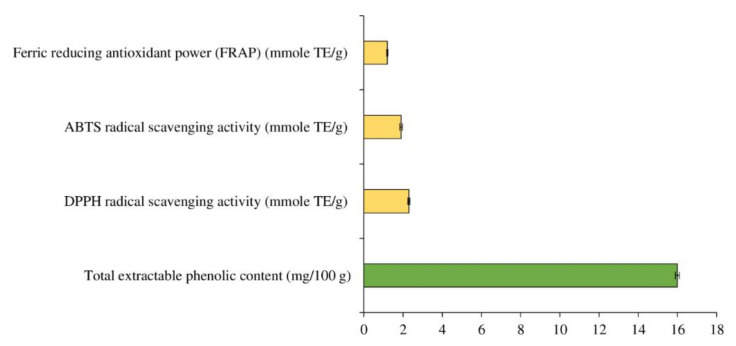
Total extractable phenolic content, DPPH radical scavenging activity, ABTS radical scavenging activity, and ferric reducing antioxidant power (FRAP) of Mon-Pu leaf extract (MPE). TE = Trolox equivalent. Bars represent the standard deviations from triplicate determinations. Different letters indicate the significant differences (*p* < 0.05).

**Figure 3 foods-11-01547-f003:**
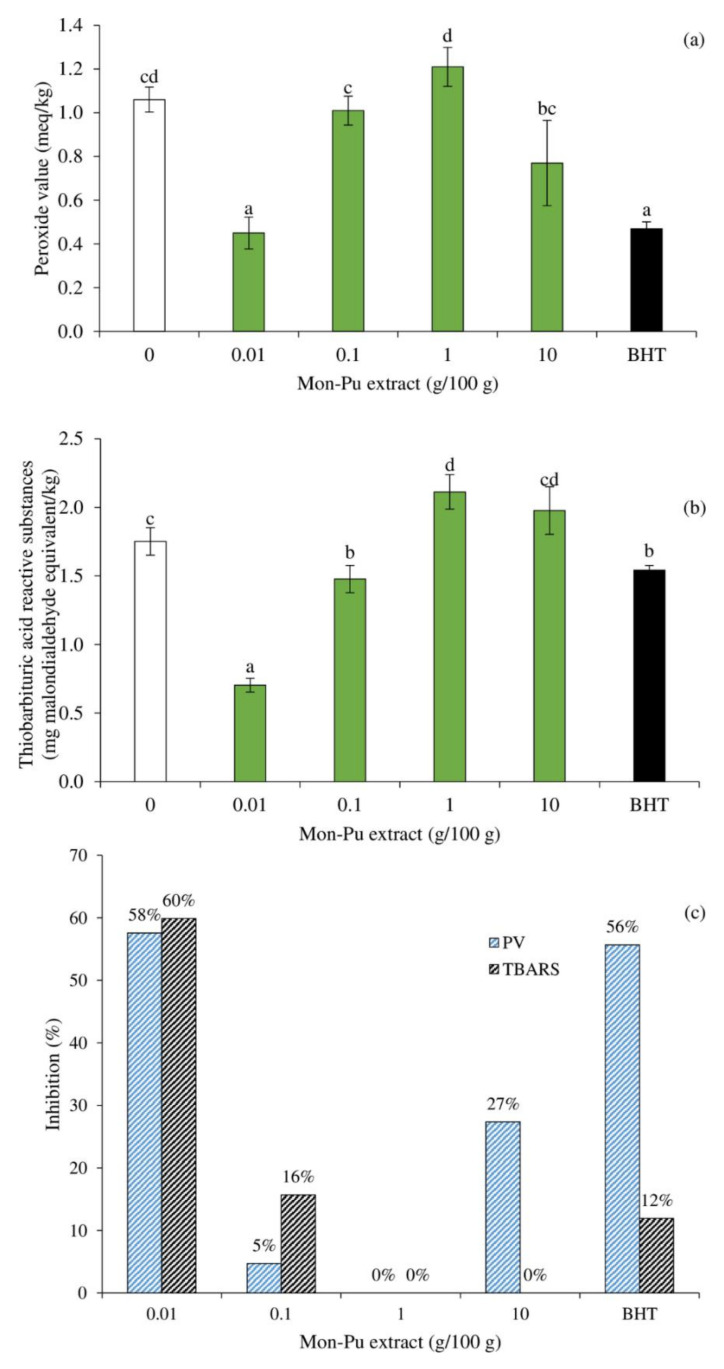
Effect of Mon-Pu leaf extract (MPE) on peroxide value (PV) (**a**), thiobarbituric acid reactive substances (TBARS) (**b**), and lipid oxidation inhibition (**c**) of a cooked sausage model system (SMS). Butylated hydroxyl toluene (BHT) at 0.003% was used to compare. The inhibitions of PV and TBARS were calculated relative to the control without MPE and BHT. Bars represent the standard deviations from triplicate determinations. Different letters indicate the significant differences (*p* < 0.05).

**Figure 4 foods-11-01547-f004:**
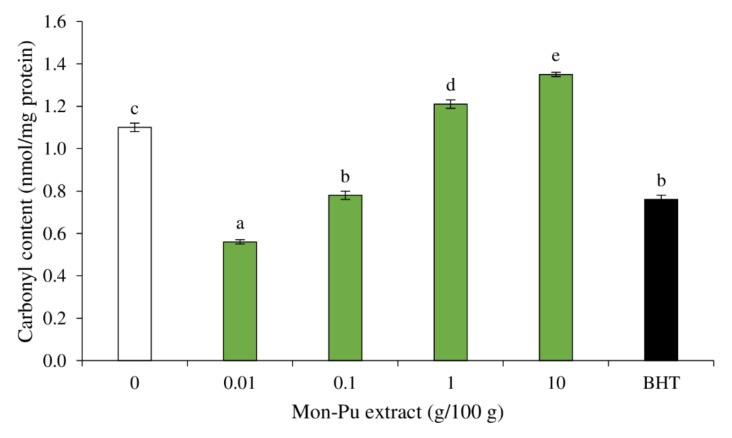
Effect of Mon-Pu leaf extract (MPE) on protein oxidation measured by carbonyl content of a cooked sausage model system (SMS). Butylated hydroxyl toluene (BHT) at 0.003% was used to compare. Bars represent the standard deviations from triplicate determinations. Different letters indicate the significant differences (*p* < 0.05).

**Figure 5 foods-11-01547-f005:**
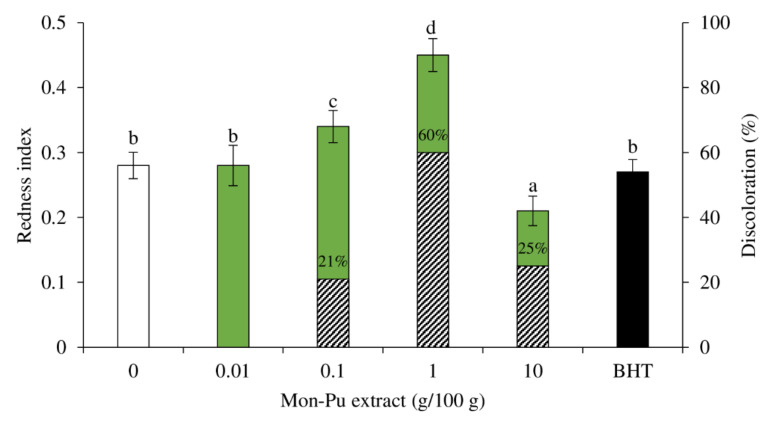
Effect of Mon-Pu leaf extract (MPE) on redness index and percentage discoloration (pattern fill) of a cooked sausage model system (SMS). Butylated hydroxyl toluene (BHT) at 0.003% was used to compare. The percentage discoloration was calculated relative to the control without MEP and BHT. Bars represent the standard deviations from triplicate determinations. Different letters indicate the significant differences (*p* < 0.05).

**Figure 6 foods-11-01547-f006:**
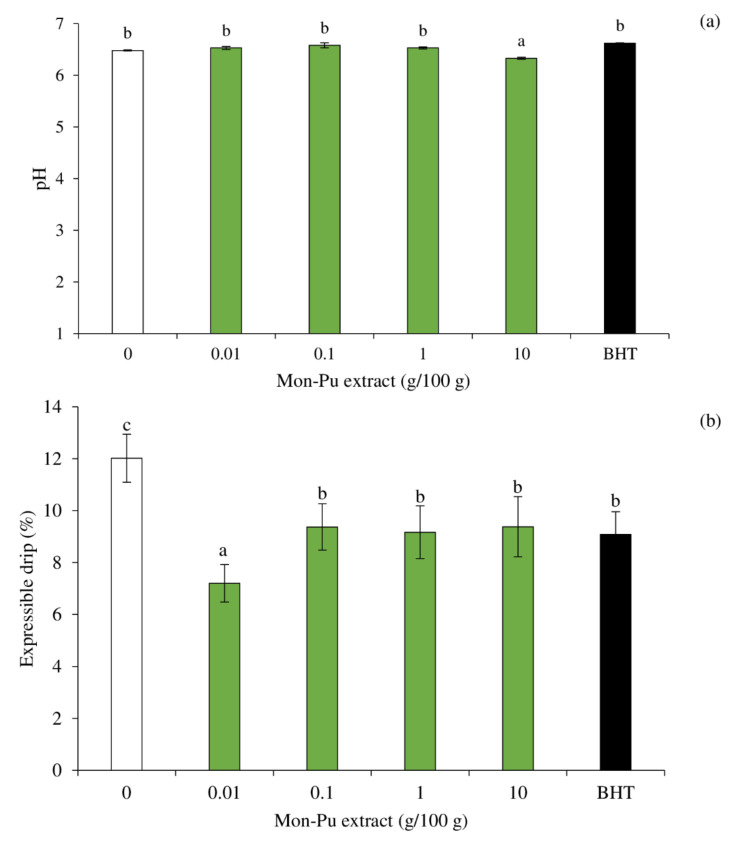
Effect of Mon-Pu leaf extract (MPE) on pH (**a**) and expressible drip (**b**) of a cooked sausage model system (SMS). Butylated hydroxyl toluene (BHT) at 0.003% was used to compare. Bars represent the standard deviations from triplicate determinations. Different letters indicate the significant differences (*p* < 0.05).

**Figure 7 foods-11-01547-f007:**
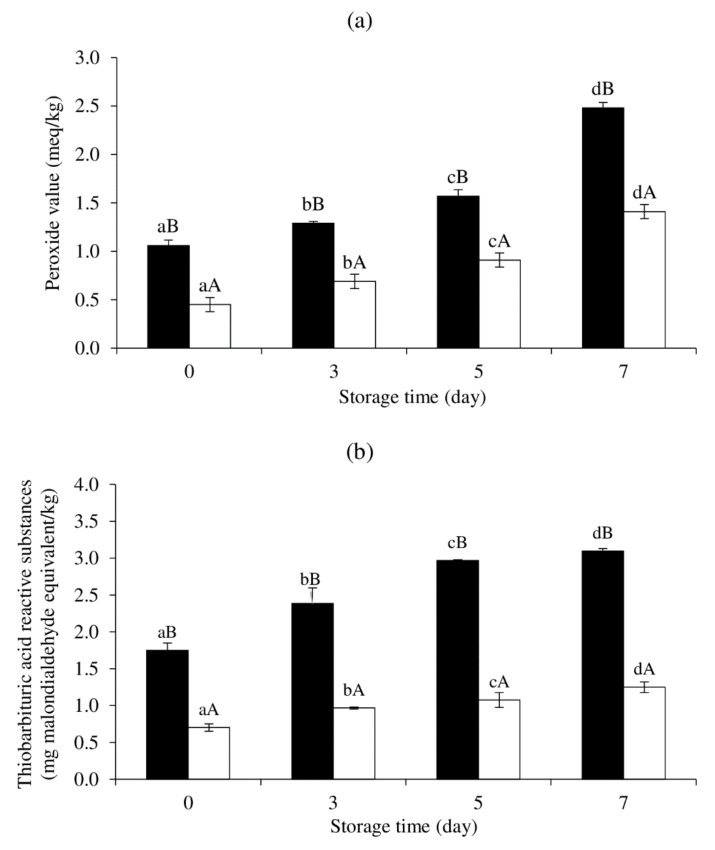
Changes in peroxide value (PV) (**a**) and thiobarbituric acid reactive substances (TBARS) (**b**) of a cooked sausage model system (SMS) added with (☐) and without (■) 0.01% (*w*/*w*) Mon-Pu leaf extract (MPE). Bars represent the standard deviations from triplicate determinations. Different lowercase letters in the same treatment and different uppercase letters at the same storage time indicate the significant differences (*p* < 0.05).

## Data Availability

The data that support the findings of this study are available from the corresponding author upon reasonable request.

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
