# Peer review of "Glochidion wallichianum Leaf Extract as a Natural Antioxidant in Sausage Model System"

_foods, 2022, doi:10.3390/foods11111547_

Round 1

Reviewer 1 Report

  • The present study evaluated the effect of Glochidion wallichianum Leaf Extract on quality characteristics of Emulsified Meat Model System. The manuscript is well written but the design is deeply flawed and warrants rejection.
  • How can someone study the effect of a plant extract on lipid stability of a meat product without doing a storage study. The authors have added an antioxidant and studied the lipid oxidative parameters on very first day and did not bother to store the product and study how it will affect the lipid oxidation over a storage period. If you want to keep a product for one day only why to add an antioxidant at all.
  • How can authors add a plant extract up to a level of 10% (too high) without doing any sensory analysis, the plant extracts at this level can negatively affect the sensory quality, we determine the level of incorporation on several criteria including sensory evaluation.
  • The product studied is not a meat emulsion, if authors wanted to study the effect of extract on emulsion, should have not filled it into casing and not cooked. The product gives an impression of some kind of a sausage.
  • I could not understand how the addition of a source of polyphenols can have a prooxidative effect and lead to an increase of peroxide value, thiobarbituric acid reactive substances and protein carbonyl contents at higher levels and decrease the same parameters at lower levels.
  • We cannot conclude and interpret anything about a plant extract if we do not do a storage study. Usually, lipid oxidative parameters do not change much during first few days despite of addition of an plant extract.
  • I recommend the authors study the effect of antioxidant for at least one week

Author Response

Reviewer 1

Comments and Suggestions for Authors

  • The present study evaluated the effect of Glochidion wallichianum Leaf Extract on quality characteristics of Emulsified Meat Model System. The manuscript is well written but the design is deeply flawed and warrants rejection.

Ans: A revision was made according to your invaluable suggestion. The storage test was also added.

  • How can someone study the effect of a plant extract on lipid stability of a meat product without doing a storage study. The authors have added an antioxidant and studied the lipid oxidative parameters on very first day and did not bother to store the product and study how it will affect the lipid oxidation over a storage period. If you want to keep a product for one day only why to add an antioxidant at all.

Ans: Actually, the storage stability was tested. The result was given in Fig. 7 and the discussion was made. Also, the Abstract, Methods, Conclusion and references were revised accordingly.

  • How can authors add a plant extract up to a level of 10% (too high) without doing any sensory analysis, the plant extracts at this level can negatively affect the sensory quality, we determine the level of incorporation on several criteria including sensory evaluation.

Ans: Thank you very much for the invaluable suggestion. Our next experiment will include a sensory test. Although the sensory acceptability of the meat product was not tested in this investigation, the discoloration was estimated in relation to the redness index of the control to observe how the plant extract affected the color of the meat product.

  • The product studied is not a meat emulsion, if authors wanted to study the effect of extract on emulsion, should have not filled it into casing and not cooked. The product gives an impression of some kind of a sausage.

Ans: Instead of "emulsified meat model system," the term "sausage model system (SMS)" was used. The modifications were implemented across the text.

  • I could not understand how the addition of a source of polyphenols can have a prooxidative effect and lead to an increase of peroxide value, thiobarbituric acid reactive substances and protein carbonyl contents at higher levels and decrease the same parameters at lower levels.

Ans: This was in agreement with several studies that the antioxidant/prooxidant activity of phenolic compounds was concentration-dependent. It was stated in Section 3.2 that “It was consistent with Cheng et al. [46], who noticed a remarkable upward trend in inhibiting lipid oxidation as mulberry polyphenolic addition increased from 0 to 1.0%, followed by a rapid decline once the addition exceeded 2.5%, with the highest lipid oxidation at 10% mulberry polyphenols added in myofibrillar protein emulsion. This could be explained that excessive polyphenols interacted with myofibrillar protein, causing a change in myofibrillar protein molecular structure at the interface. A high concentration of phenolics may also serve as a deterrent, preventing the formation of a continuous protein layer around oil droplets, thereby preventing the formation of high-elastic emulsion [46]. The high dose of phenolic compounds promotes the production of amino-quinone and thiol-quinone adducts, which may prevent intermolecular interactions from forming between proteins at the emulsion interface. The disordered emulsion may also be able to react with atmospheric oxygen, stimulating lipid oxidation in SMS treated with high MPE concentrations. Aside from antioxidative activity, certain phenolic excipients may actively contribute to protein unfolding and denaturation, leading to a significant reduction in the antioxidant capacity of the MPE in SMS [47].”

In section 3.3., it was stated that “It should be noted that the addition of 1-10% MPE to SMS resulted in higher protein carbonyl content than the control sample (p < 0.05), indicating that high amounts of MPE stimulate protein oxidation. It was consistent with Cando et al. [14], who investigated the significant pro-oxidant effect of Willowherb extract when added to beef patties at high concentrations (800 ppm). The pro-oxidative functions of polyphenols have been attributed to their activity to convert molecular oxygen to ROS and Fe3+ to Fe2+ [50]. The increase in ROS formation and/or support to the iron oxidizing cycle may theoretically influence both lipids and proteins. As a result, the increase in protein carbonylation mediated by MPE phenolics could be explained by the particular interaction mechanisms between phenolics and specific protein residues in dietary proteins [49]. According to this pathway, the δ-amino group of alkaline amino acids (lysine, arginine, and/or proline) can interact with quinone derivatives of phenolics formed in the presence of transition metals, initiating the oxidative deamination of the amino acid and eventually generating the respective carbonyl compound. [49]. It is thus possible that an undetermined proportion of the phenolics in the MPE were oxidized in the presence of endogenous iron, and the resulting quinones catalyzed the formation of protein carbonyls due to their deamination activity on alkaline amino acids [51]. As a result, the quinone forms would have aided oxidative deamination of alkaline amino acids, as well as lipid oxidation.”

Also, in Section 3.3, it was stated that “Plant phenolics, as redox-active compounds, have been reported to have both pro-oxidant and antioxidant effects on biological systems [32], relying on the availability of other redox-active substances and transition metals [50].

  • We cannot conclude and interpret anything about a plant extract if we do not do a storage study. Usually, lipid oxidative parameters do not change much during first few days despite of addition of an plant extract.

Ans: The storage test was completed, however the results were not included in the initial submission. It was added and discussed in Fig. 7 in a revised version.

  • I recommend the authors study the effect of antioxidant for at least one week

Ans: As mentioned above, storage stability was examined, as indicated in Fig. 7.  The Abstract, Methods, Conclusion, and references were also updated to reflect this.

Reviewer 2 Report

Dear Authors,

please see the review of your manuscript I have attached below.

Review of the manuscript

Manuscript ID: foods-1742605

Title: Glochidion wallichianum Leaf Extract as a Natural Antioxidant in Emulsified Meat Model System

A brief summary

The aim of the research was to evaluate the bioactive properties of an 80% ethanolic extract from Glochidion wallichianum (Thai name: Mon-Pu) and its effect on the oxidative stability and physicochemical properties of cooked emulsified meat model system. The antioxidant activity of the ethanolic extract of Glochidion wallichianum was assessed with the use of three different methods, i.e. with the DPPH radical, with the ABTS radical and as ferric reducing antioxidant power. Plant extract was added to the meat system at a different levels. The effect of its addition on lipid oxidation, protein oxidation, and discoloration was assessed and compared to synthetic butylated hydroxyl toluene (BHT) and a control (without antioxidant). Based on the result of such measurements as: peroxide value, thiobarbituric acid reactive substances, and protein carbonyl contents it was found that the addition of Glochidion wallichianum extract had a prooxidative effect in meat system. Another negative effect of using plant extract was that its incorporation into emulsified meat system at higher concentrations (1-10%) led to the discoloration of meat system. Taking into account all results obtained, the authors concluded that in order to obtain satisfactory results, i.e. to slow down the processes of lipid and protein oxidation, to minimize the impact on the technological properties of the emulsion-based meat product, and to limit undesirable color changes, the authors recommend the use of a small amount of plant extract in a formulation (i.e. about 0.01% (w/w). A supposition has been made, that ethanolic extract from Glochidion wallichianum could than serve as a good replacer of role the synthetic food additive BHT.

General comments

After reading the manuscript I have stated that it is written in plain English what makes it clear. (In my opinion the English language is appropriate and understandable). The manuscript is relevant for the field and presented in a well-structured manner.

The manuscript is scientifically sound and represents the experimental design appropriate to test the hypothesis. The test methods were described in such a way that the tests could be repeated. The conclusions were based on the measurement results.

In the article Authors assessed the antioxidant properties of the plant Glochidion wallichianum, commonly known in Asia as Mon-Pu. An element of the novelty may be the evaluation of the possibility of using the plant extract in order to improve the antioxidant stability of the model meat product.

The manuscript fits in the journal scope.

The ethics and data availability statements have not been included (not applicable).

The work expands the current knowledge in the field of the possibility of using plant preparations in meat processing, including the benefits of technological and qualitative nature.

The advantage of the article is the exhaustive description of the plant Glochidion wallichianum, growing in Southeast Asia. From the point of view of potential application of Glochidion wallichianum to food, it is important that biologically active compounds are presented that are present in this plant and could be used as inhibitors of oxidation processes. In my opinion, the value of the obtained results would be greater if the sensory acceptance of the color and the palatability of the model meat product were assessed. - Are such studies planned?

Specific comments

Lines 49-55: The Authors have stated that ‘the synthetic phenolic antioxidants butylated hydroxyanisole (BHA) and butylated hydroxy toluene (BHT) are commonly used in uncured meats’. Such a statement is imprecise. Please note that in Europe (European Union members) BHA and BHT are not widely used in the production of processed meats! Their use is allowed only in special cases. They are listed in COMMISSION REGULATION (EU) No 1129/2011 (please see Annex II, AUTHORISED FOOD ADDITIVES AND CONDITIONS OF USE IN FOOD CATEGORIES). I believe that a reference to European legislation on the use of synthetic antioxidants in meat products would improve the article and increase the interest of a larger group of potential readers.

Lines 151-160: Was the temperature inside the model meat product measured? Does the applied heat treatment temperature ensure health safety?

Lines 175-182: In the opinion of the authors, should it not be necessary to check whether the colour of the model meat product was sensory acceptable? Did the addition of plant (Glochidion wallichianum) extract deteriorate the colour of the meat product?

Regards,

Reviewer

Author Response

Reviewer 2

Dear Authors,

please see the review of your manuscript I have attached below.

Review of the manuscript

Manuscript ID: foods-1742605

Title: Glochidion wallichianum Leaf Extract as a Natural Antioxidant in Emulsified Meat Model System

A brief summary

The aim of the research was to evaluate the bioactive properties of an 80% ethanolic extract from Glochidion wallichianum (Thai name: Mon-Pu) and its effect on the oxidative stability and physicochemical properties of cooked emulsified meat model system. The antioxidant activity of the ethanolic extract of Glochidion wallichianum was assessed with the use of three different methods, i.e. with the DPPH radical, with the ABTS radical and as ferric reducing antioxidant power. Plant extract was added to the meat system at a different levels. The effect of its addition on lipid oxidation, protein oxidation, and discoloration was assessed and compared to synthetic butylated hydroxyl toluene (BHT) and a control (without antioxidant). Based on the result of such measurements as: peroxide value, thiobarbituric acid reactive substances, and protein carbonyl contents it was found that the addition of Glochidion wallichianum extract had a prooxidative effect in meat system. Another negative effect of using plant extract was that its incorporation into emulsified meat system at higher concentrations (1-10%) led to the discoloration of meat system. Taking into account all results obtained, the authors concluded that in order to obtain satisfactory results, i.e. to slow down the processes of lipid and protein oxidation, to minimize the impact on the technological properties of the emulsion-based meat product, and to limit undesirable color changes, the authors recommend the use of a small amount of plant extract in a formulation (i.e. about 0.01% (w/w). A supposition has been made, that ethanolic extract from Glochidion wallichianum could than serve as a good replacer of role the synthetic food additive BHT.

General comments

After reading the manuscript I have stated that it is written in plain English what makes it clear. (In my opinion the English language is appropriate and understandable). The manuscript is relevant for the field and presented in a well-structured manner.

The manuscript is scientifically sound and represents the experimental design appropriate to test the hypothesis. The test methods were described in such a way that the tests could be repeated. The conclusions were based on the measurement results.

In the article Authors assessed the antioxidant properties of the plant Glochidion wallichianum, commonly known in Asia as Mon-Pu. An element of the novelty may be the evaluation of the possibility of using the plant extract in order to improve the antioxidant stability of the model meat product.

The manuscript fits in the journal scope.

The ethics and data availability statements have not been included (not applicable).

The work expands the current knowledge in the field of the possibility of using plant preparations in meat processing, including the benefits of technological and qualitative nature.

The advantage of the article is the exhaustive description of the plant Glochidion wallichianum, growing in Southeast Asia. From the point of view of potential application of Glochidion wallichianum to food, it is important that biologically active compounds are presented that are present in this plant and could be used as inhibitors of oxidation processes. In my opinion, the value of the obtained results would be greater if the sensory acceptance of the color and the palatability of the model meat product were assessed. - Are such studies planned?

Ans: Thank you very much. Our next experiment will include sensory analysis. Chemical and physicochemical parameters were examined in this study.

Specific comments

Lines 49-55: The Authors have stated that ‘the synthetic phenolic antioxidants butylated hydroxyanisole (BHA) and butylated hydroxy toluene (BHT) are commonly used in uncured meats’. Such a statement is imprecise. Please note that in Europe (European Union members) BHA and BHT are not widely used in the production of processed meats! Their use is allowed only in special cases. They are listed in COMMISSION REGULATION (EU) No 1129/2011 (please see Annex II, AUTHORISED FOOD ADDITIVES AND CONDITIONS OF USE IN FOOD CATEGORIES). I believe that a reference to European legislation on the use of synthetic antioxidants in meat products would improve the article and increase the interest of a larger group of potential readers.

Ans: Thank you very much for your invaluable suggestion. It was changed to “Cured meats contain sodium nitrite, which is a very effective antioxidant whereas, in some countries, the synthetic phenolic antioxidants butylated hydroxy anisole (BHA) and butylated hydroxy toluene (BHT) are commonly used in uncured meats [6, 11]. According to USDA guidelines, BHA and BHT are allowed up to 0.01% (based on fat content) in fresh sausage and 0.003% (based on total weight) in dry sausage [6]. According to Commission Regulation (EU) No 1129/2011 [10], antioxidants such as ascorbic acid and its salts, phosphoric acid, calcium disodium ethylene diamine tetraacetate (calcium disodium EDTA), and rosemary extracts are permissible in heat-treated processed meat in European Union countries. However, because synthetic antioxidants are causing worry among consumers, numerous meat processors are exploring for natural antioxidant alternatives.

The reference was updated. “10. EU Commission. Commission Regulation (EU) No 1129/2011 of 11 November 2011 amending Annex II to Regulation (EC) No 1333/2008 of the European Parliament and of the Council by establishing a Union list of food additives. Off. J. Eur. Union L. 2011, 295, 1-177.

Lines 151-160: Was the temperature inside the model meat product measured? Does the applied heat treatment temperature ensure health safety?

Ans: The temperature inside the product was measured and it was not less than 72 °C to ensure the food safety. The statement was added “After stuffing into the casing (2-cm diameter), two-step cooking (60 °C/30 min followed by 90 °C/30 min) was applied. The product's core temperature was not less than 72 °C to ensure food safety.

Lines 175-182: In the opinion of the authors, should it not be necessary to check whether the colour of the model meat product was sensory acceptable? Did the addition of plant (Glochidion wallichianum) extract deteriorate the colour of the meat product?

Ans: In this study, the instrumental color parameter was measured and reported as redness index. The sensory acceptable was not determined in this study. However, the discoloration was estimated relatively to the redness index of the control to see the effect of plant extract on the color of the meat product. This was discussed in Section 3.4.

Round 2

Reviewer 1 Report

The authors have revised the manuscript and addressed all my concerns.